# LncRNA 220: A Novel Long Non-Coding RNA Regulates Autophagy and Apoptosis in Kupffer Cells via the miR-5101/PI3K/AKT/mTOR Axis in LPS-Induced Endotoxemic Liver Injury in Mice

**DOI:** 10.3390/ijms241311210

**Published:** 2023-07-07

**Authors:** Ying Yang, Tian Tian, Shan Li, Nanhong Li, Haihua Luo, Yong Jiang

**Affiliations:** Guangdong Provincial Key Laboratory of Proteomics, State Key Laboratory of Organ Failure Research, Department of Pathophysiology, School of Basic Medical Sciences, Southern Medical University, Guangzhou 510515, China; zxcvb000@i.smu.edu.cn (Y.Y.); tiantian321@i.smu.edu.cn (T.T.); lishan19900325@i.smu.edu.cn (S.L.); lnh13751330113@i.smu.edu.cn (N.L.); btxlhh@smu.edu.cn (H.L.)

**Keywords:** sepsis, acute liver injury, long non-coding RNA, microRNA, competing endogenous RNA, autophagy, apoptosis

## Abstract

Sepsis is a severe medical condition distinguished by immune systematic dysfunction and multiple organic injury, or even failure, resulting from an acute systemic inflammatory response. Acute liver injury (ALI) could be considered as a notable inflammatory outcome of sepsis. Studies have demonstrated the essential roles played by long non-coding RNAs (lncRNAs) in mediating the processes of various diseases, including their ability to engage in interactions with microRNAs (miRNAs) as complexes of competing endogenous RNA (ceRNA) to modulate signaling pathways. In this study, a newly discovered lncRNA, named 220, was identified to function in regulating autophagy and apoptosis in Kupffer cells treated with lipopolysaccharide (LPS). This was achieved through sponging miR-5101 as a ceRNA complex, as identified via high-throughput sequencing. The expression of 220 was found to be significantly different in the hepatic tissues of endotoxemic mice that were treated with LPS for 8 h, ultimately modulating the ALI process. Our studies have collectively demonstrated that 220 is a novel regulator that acts on LPS-induced autophagy and apoptosis in Kupffer cells, thereby mediating the ALI process induced by LPS. Furthermore, the validation of our findings using clinical databases suggests that 220 could potentially serve as a molecular target of clinical, diagnostic, and therapeutic significance in septic liver injury.

## 1. Introduction

Sepsis, a frequently fatal syndrome triggered by severe infections, could contribute to multiple organ dysfunction syndrome (MODS) [1,2]. The significant mortality rate associated with sepsis underscores its status as a significant issue in the realm of public health [3,4]. The pathogenesis of sepsis involves a complex interplay of cellular and molecular events that ultimately culminate in multiple organic failure, resulting in a critical illness characterized by severe dysimmunity and metabolic disturbance [5]. The principle of septic therapy encompasses fluid resuscitation, hemodynamic support, the use of antibiotics, and source control [6]. Despite advancements in antibiotics and novel therapies, the mortality rates associated with septic shock persist at an unacceptably elevated level, and the prevalence of this ailment among patients admitted to hospitals has increased by almost two-fold over the same time frame [7,8]. To find effective therapeutic approaches, research on the mechanisms of the expeditious onset and advancement of sepsis must be considered as matters of urgency.

The liver plays a pivotal role in modulating the immune response to pathogens or other stimuli [9]. Acute liver injury (ALI), a significant complication of sepsis, exacerbates syndromes and leads to unfavorable prognoses [10]. The emergence of ALI in sepsis can be ascribed to both hemodynamic compromise and the direct activation of monocytes, macrophages, and hepatocytes. The physiological reaction characterized by inflammation in sepsis is primarily mediated by activating the pathogen recognition receptors, such as Toll-like receptors (TLRs), in monocytes and macrophages (including hepatic Kupffer cells) that are essential to the pathogenesis of septic liver injury [11,12]. Kupffer cells, regarded as essential constituents of the mononuclear phagocytic system, play a crucial role in the hepatic and systemic reaction to pathogens, indicating their potential as crucial mediators of liver injury and subsequent restoration [13]. The activation of Kupffer cells is imperative for the liver’s reaction to infection or injury, as they may safeguard the liver from infection during the ensuing inflammatory response, thereby curtailing cellular and organic harm to the host [14]. Toll-like receptor 4 (TLR4) is a fundamental pattern recognition receptor of monocytes and macrophages in the innate immune system, given its ability to detect the presence of lipopolysaccharides (LPS) [15,16,17]. Several studies have revealed that the activation of TLR4 signaling may participate in the protective mechanism of inflammation [18]. The activation of nuclear factor-kappa B (NF-κB) and interferon regulatory factors (IRFs) by TLR4 is dependent on the adaptor proteins myeloid differentiation factor 88 (MyD88) and Toll/interleukin-1 receptor-domain-containing adaptor-inducing interferon-β (TRIF) [19]. However, the TLR4/MyD88-dependent and TLR4/TRIF-dependent pathways play significant roles in the escalation of inflammation in liver disease [20,21]. Activation of the MyD88/TRIF/NF-κB pathway in Kupffer cells via TLR-mediated signaling results in the generation of substantial quantities of pro-inflammatory cytokines, such as tumor necrosis factor (TNF-α), interleukin 1β (IL-1β), and interleukin 6 (IL-6) [20]. This, in turn, triggers the activation of the innate immune system and results in a state of hyperinflammation. On the contrary, the stimulation of Kupffer cells and hepatocytes via TNF-α and IL-1β leads to the synthesis of IL-6, which subsequently triggers the synthesis of acute-phase proteins by activating signal transducer and activator of transcription 3 (STAT3) [22]. Extensive research has demonstrated that TNF-α, IL-1β, and IL-6 are pivotal in modulating the initial response of the innate immune system to impairment or infection [23]. These three cytokines are primarily accountable for the manifestation of systemic inflammatory response syndrome (SIRS) and also have the potential to serve as biomarkers for sepsis [20,24]. The generation of TNF-α, IL-1β, and IL-6 during the progression of ALI holds significant importance, and the interplay between activated Kupffer cells and hepatocytes may serve as a pivotal element in the pathogenesis of ALI [22]. The overexpression of these three cytokines during ALI can trigger Kupffer cell dysfunction, leading to cellular apoptosis and autophagy in severe cases [23]. Nevertheless, the excessive apoptosis and autophagy of Kupffer cells can result in their decreased survival during the inflammatory phase, ultimately culminating in ALI. This phenomenon also warrants serious consideration.

Long non-coding RNAs (LncRNAs), which are RNA molecules exceeding 200 nucleotides in length and devoid of coding functions, have been extensively investigated due to their significant involvement in inflammatory responses. These RNAs can be categorized into two types based on their subcellular localization: nuclear lncRNAs and cytoplasmic lncRNAs. Nuclear lncRNAs primarily regulate transcriptional factor activity in gene transcription, participate in chromatin remodeling and nuclear reconstruction, and mediate mRNA stability. In contrast, cytoplasmic lncRNAs function mainly in sequestering miRNAs as a ceRNA complex to modulate gene expression, regulating the alternative splicing of mRNA and mediating protein–protein interactions [25]. Prior research has revealed that numerous long non-coding RNAs (lncRNAs) are situated within the nucleus [26]. However, when exposed to external stimuli such as LPS, they are capable of executing nucleocytoplasmic shuttling [27].

MicroRNAs (MiRNAs), which are a substantial class of RNA molecules without coding functions, have been observed to target mRNA to form RNA-induced silencing complexes (RISC), thereby contributing to cell survival and organ development [28,29,30,31]. Extensive research has demonstrated that lncRNA can act as a competing endogenous RNA (ceRNA) complex to modulate post-transcriptional gene processes [32]. For instance, pertinent research has revealed that lncRNA taurine-upregulated gene 1 (TUG1) and silencing information regulator 1 (SIRT1) possess complementary binding sites with miR-200a-3p, indicating a potential association between TUG1 or SIRT1 and miR-200a-3p in the context of sepsis-induced hepatic injury [33]. Additionally, the differential expression of lncRNA colorectal neoplasia differentially expressed (CRNDE) in combination with miR-126-5p has been shown to alleviate sepsis-induced liver injury [34]. Consequently, both lncRNA and miRNA hold promise as potential diagnostic biomarkers or therapeutic targets for hepatic diseases.

The phosphatidylinositol-3-kinase (PI3K/AKT) and mammalian target of rapamycin (mTOR) signaling pathways, which are deemed critical to cellular outcomes in pathophysiological states, may be considered as principal regulators of cell survival in the presence of inflammatory conditions [35]. Furthermore, the activation of the PI3K/AKT/mTOR pathway can facilitate cell proliferation and death, thereby modulating inflammation and liver injury in septic endotoxemia [36,37].

Autophagy plays an essential role in preserving cellular integrity during periods of stress by facilitating the degradation of cellular components [38]. Impaired autophagy has been observed in autoimmune diseases and other inflammatory conditions [39]. Autophagosomes are widely recognized as the hallmark of autophagy, given their pivotal role in the breakdown of intracellular constituents. In order to detect autophagic flux, it is imperative to observe the proteins implicated in the formation of autophagosomes [40]. Facilitated by the microtubule-associated protein 1 light chain 3 B (LC3B), the cytosolic form of LC3B-Ⅰ can undergo transformation into the form of LC3B-Ⅱ that has undergone lipidation during an elevation in autophagic flux, suggesting that the transformation of cytosolic LC3B-Ⅰ into lipidated LC3B-Ⅱ may serve as a crucial marker for autophagy. Furthermore, sequestosome 1 (SQSTM1) can direct ubiquitinated substrates to autophagosomes by interacting with LC3B, making these two proteins potential biomarkers for assessing autophagic flux [41]. The induction of autophagy is governed by the Unc-51-like kinase (ULK) complex, which is suppressed by mTOR. Additionally, the PI3K/AKT pathway also exerts a significant regulatory control over this process [42,43]. However, excessive autophagy, in certain circumstances, could result in the uncontrolled degradation of cellular components and subsequent cell death. This phenomenon has been termed autosis and continues to be a subject of research significance in various diseases [44].

Apoptosis is a vital process that facilitates the systematic and effective elimination of damaged cells [45]. The participation of caspases is indispensable to the mechanism of apoptosis, as they function as both initiators (caspase-3, -8, -9, and -10) that activate the apoptotic pathway and executors (caspase-3, -6, and -7) that facilitate the degradation of cellular constituents [46]. Furthermore, the crucial involvement of poly (ADP-ribose) polymerase (PARP) in DNA damage is attributed to its status as the substate of activated caspases [47]. Nevertheless, the activation of caspase-3, -6, and -7, which are accountable for the distinctive morphological alterations observed during apoptosis, is the ultimate outcome of both the extrinsic and intrinsic apoptotic pathways [48]. These proteins may therefore serve as significant metrics in the investigation of cell apoptosis. AKT-activated mTOR, which is initiated by PI3K, has been shown to induce apoptosis [49,50]. Notably, excessive apoptosis can lead to uncontrolled cell death and organ injury in various diseases [51], highlighting the need for further attention to this phenomenon.

SIRS is initially modeled by the immune response to endotoxin, a type of LPS found within Gram-negative bacteria’s cell walls [52]. Endotoxin represents a prototypical pathogen-associated molecular pattern (PAMP) that can be recognized by corresponding receptors in innate immune cells, including Kupffer cells [53]. Indeed, LPS-induced SIRS in mice is widely accepted as a valuable model for investigating the pathophysiological mechanisms of inflammation in sepsis, given its ability to mimic key pathological events, such as inflammatory and histological alterations witnessed in this ailment [54,55]. The objective of this investigation was to examine the correlation between lncRNAs and miRNAs in the occurrence and development of sepsis-associated ALI. To achieve this, systemic inflammated animal models induced by LPS and inflammatory cell models were constructed using C57 mice and murine Kupffer cells, respectively. A novel lncRNA, named 220, based on its chromosomal location, was identified. This investigation explored the potential role of 220 as a molecular sponge of miR-5101 in the autophagic and apoptotic processes of Kupffer cells via the PI3K/AKT/mTOR axis during inflammatory situations. Furthermore, numerous homologous sequences of 220 on human chromosome 21 were identified via the NCBI database, and their clinical significance was validated by screening the corresponding dataset of septic patients in the GEO database. The regulatory function of 220, as demonstrated in our study through laboratory modeling and clinical database analysis, establishes an experimental foundation for its potential use as a novel target of clinical, diagnostic, and therapeutic significance in sepsis.

## 2. Results

### 2.1. Identification and Annotation of lncRNA 220

Systemic inflammated animal models induced by LPS were established in C57 mice aged 8–12 weeks through the administration of a single dose of 20 mg/kg of LPS for treatment. The H&E results revealed the presence of LPS-induced ALI, which was significantly more pronounced in comparison with the control group (Figure 1A(a)). Notably, the hepatic tissues exhibited a substantial amount of spotty necrosis, congestion in the central vein, disorganized permutation in hepatocytes (Figure 1A(b)), and small focal necrosis (Figure 1A(c,d)). As stated in the introduction, the pathogenesis of ALI is linked to the interplay between activated Kupffer cells and hepatocytes, with this interaction serving as a key element. Additionally, the inflammatory apoptosis and autophagy mechanisms are critical for cell survival and contribute to the development of ALI by modulating this cellular interplay [56,57].

In order to further investigate the regulatory role of lncRNAs in the ALI process, systemic inflammated animal models were utilized for RNA sequencing (RNA-seq). The differential expression of lncRNAs (DELs) was screened using volcanic plots based on the RNA-seq results (Figure 1B). The findings revealed that 40 DELs were identified following LPS treatment for 2 h, while 87 and 12 DELs were identified after 8 and 24 h of LPS treatment, respectively. These results suggested that lncRNAs were primarily expressed during the 8 h period. Prior research has established that ALI may manifest at any point during sepsis, indicating its potential significance as a marker for MODS [58]. Collectively, a duration of 8 h of LPS treatment may serve as the intervention condition in the present investigation. The heat map was utilized to demonstrate the distinct expressions of lncRNAs at various intervening time points, which yielded congruent outcomes with the volcanic plots (Appendix A). Consequently, the differentially expressed lncRNAs observed during 8 h of treatment with LPS could be considered as the primary focus of this study. Employing the screening criteria of FDR < 0.05 and | log2 (Fold change) | > 1, a heat map was generated to showcase the top 30 differentially expressed lncRNAs (Figure 1C). Based on these findings, lncRNA TCONS_00127483 was identified as the most significantly differentially expressed lncRNA following LPS treatment for 8 h (Figure 1D). The UCSC database facilitated the identification of a newly discovered lncRNA, namely lncRNA TCONS_00127483, situated on the chromosome 11q strand in mice. This lncRNA spans a length of 1888-nt and comprises two exons and one intron (Figure 1E). The Appendix A provide access to the sequence of this lncRNA. Following its location-based nomenclature, lncRNA TCONS_00127483 was renamed as lncRNA 220, abbreviated as 220.

The hepatic tissues of mice (*n* = 5 per group) were analyzed for the expression of 220 treated with LPS at different time points (0 h, 2 h, 4 h, 6 h, 8 h, 12 h, and 24 h) to validate the RNA-seq results (Figure 1F). As outlined in the introduction, the TLR4/MyD88-dependent and TLR4/TRIF-dependent pathways are crucial contributors to increased inflammation during LPS-associated liver disease. The MyD88/TRIF/NF-κB pathway, which is considered the primary regulatory downstream pathway of TLR4, mediates various inflammatory processes during LPS-induced infection. To further investigate this pathway, TLR4^−/−^, MyD88^−/−^, and TRIF^−/−^ mice were utilized. The qPCR findings evinced that the expression of 220 was repressed in TLR4^−/−^ and MyD88^−/−^ mice (Figure 1G,H), thereby suggesting that TLR4 and MyD88 served as the upstream regulators of 220.

The localization of 220 was assessed via RNA-FISH at both the tissue and cellular levels. Remarkably, the FISH outcomes at the tissue level revealed that 220 was distributed in both the inflammatory and parenchymal cells (indicated by the arrows) in the liver (Figure 2A). To further specify the cell type employed in this investigation, AML-12 cells and Kupffer cells were chosen for the construction of the LPS-induced inflammatory cell models at different time points (0 h, 2 h, 4 h, 8 h, 12 h, and 24 h). The findings indicated that Kupffer cells expressed 220 more prominently than parenchymal cells in the liver (Figure 2B), suggesting that Kupffer cells may serve as in vitro models for LPS-induced inflammation. To confirm the subcellular localization of 220 in Kupffer cells, the lncLocator 2.0 was utilized, using the sequence of 220. The algorithm of this tool assigned a positive score for cytoplasmic location and a negative score for nuclear location (Appendix A), revealing that 220 was primarily located in the nucleus. To validate this prediction, the nucleocytoplasmic RNA separation assay was utilized to investigate the subcellular localization of 220 in Kupffer cells. The results indicated that, under normal conditions, 220 was predominantly located in the nucleus, which corroborated the previously hypothesized localization (Figure 2C). Following an 8-h LPS treatment, the expression of 220 in the cytoplasm significantly increased, while no significant change was observed in the nucleus (Figure 2D,E). Furthermore, the FISH analysis at the cellular level demonstrated that 220 was expressed in both the nucleus and cytoplasm, with a majority of the expression in the nucleus, as quantitated by Image J (Figure 2F,G). As previously noted in the introduction, lncRNAs are predominantly situated within the nucleus. However, when confronted with an external stimulus, they possess the capability to function as transporters between the nucleus and cytoplasm, thereby exerting corresponding regulatory functions.

Collectively, it is reasonable to hypothesize that 220 serves as a regulator in either the nucleus or cytoplasm.

### 2.2. Prediction for the Interaction between lncRNA 220 and miRNA 5101 as a ceRNA Complex

Based on the aforementioned findings, the upregulation of 220 in the cytoplasm of Kupffer cells following an 8-h LPS treatment indicates its potential role as a cytoplasmic regulator in LPS-associated inflammation. Previous studies have demonstrated that lncRNAs can modulate mRNAs through ceRNA networks [59]. The results of the nucleocytoplasmic RNA separation assay also suggested that 220 may function in the cytoplasm by interacting with miRNAs. The targeted miRNAs of 220 were identified using the miRDB database, with miR-5101, miR-669b-5p, and miR-7234-5p being deemed the most relevant (Figure 3A). The miRDB database was employed to predict the targeted mRNAs of these miRNAs (Figure 3B), and the STRING database, in conjunction with Cytoscape, was utilized to visualize the association between these mRNAs (Figure 3C). Subsequently, the Cytohubba plugin was utilized to screen for hub genes under the Degree algorithm (Figure 3D) [60], based on these findings. The functional pathway of the hub genes was determined through the application of GO/KEGG enrichment analysis, revealing their primary involvement in the PI3K signaling pathway, which mediates cell survival (Figure 3E). The ceRNA network comprising 220 and its targeted miRNAs was visualized using the Sankey plot, based on the Target Score generated by the miRDB database (Figure 3F). Here, it was demonstrated that the ordinate axis depicted three distinct variables, specifically lncRNA, miRNA, and mRNA. The horizontal axis corresponded to the values of each variable, signifying individual lncRNA, miRNA, and mRNA entities. The width of the connecting strip between variables indicated the degree of correlation between them. Notably, a distinct association was observed between 220 and miR-5101, while miR-5101 exhibited a close relationship with the PI3K complex.

Taken together, it is reasonable to hypothesize that 220 acts as a ceRNA complex by sequestering miR-5101 (abbreviated as 5101) to facilitate cell survival via the PI3K signaling pathway in the context of LPS-induced inflammation (Figure 3G).

### 2.3. Decoy of miRNA 5101 by lncRNA 220 as a ceRNA Complex

The expression of 5101 was observed to be reduced in systemic inflammated animal models that were treated with LPS for a duration of 8 h (Figure 4A). To validate the binding between 5101 and 220, both RNA pull-down (Figure 4B) and dual-luciferase reporter (Figure 4C) assays were employed. The luciferase activity was found to be inhibited upon co-transfection of Kupffer cells with Luc-220-WT plasmid and mimics of 5101 in comparison with its negative control. However, no significant difference in the luciferase activity was witnessed upon co-transfection of Kupffer cells with Luc-220-MUT plasmid and mimics of 5101 in comparison with its negative control. In order to further validate the correlation between 220 and 5101, it was imperative that the inhibitory efficacy of 220 surpassed 70% (Appendix A and Figure 4D). Notably, no noteworthy dissimilarity was observed in the expression of 5101 subsequent to the knockdown of 220 in comparison with its control group, thereby indicating a robust interaction between 220 and 5101. To further substantiate the regulatory role of 5101, both overexpression and interference techniques were employed, and their respective efficacies were assessed using qRT-PCR (Figure 4F,G). The expression level of 220 was found to be suppressed upon overexpression of 5101, while its knockdown led to a reversal of this effect (Figure 4H,I), indicating that 5101 inhibited the expression of 220. Moreover, the upregulation of TNF-α, IL-1β, and IL-6 mRNA upon overexpression of 5101 suggested its involvement in the release of inflammatory cytokines. In contrast, the Pik3ca mRNA’s expression level was suppressed by 5101, but this effect was reversed upon knockdown of Pik3ca, suggesting the existence of an interaction between 5101 and Pik3ca (Figure 4J).

Collectively, these findings demonstrate that 220 has the potential to operate as a ceRNA complex in combination with 5101, thereby facilitating interaction with Pik3ca and ultimately regulating its downstream signaling cascade.

### 2.4. Regulation of miRNA 5101 on LPS-Induced Autophagy in Kupffer Cells via the PI3K/AKT/mTOR Axis

Western blotting (WB) was utilized to assess the autophagic levels of cells subsequent to the overexpression and knockdown of 5101 at the protein level (Figure 5A–D). The results indicated that no significant difference was observed in the protein levels of PIK3CA following the overexpression and knockdown of 5101. However, the phosphorylated level of PIK3R1 was observed to increase upon overexpression of 5101, while its knockdown resulted in the reversal of this outcome. Additionally, 5101 was found to suppress the phosphorylated level of AKT, which could be reversed by its knockdown. No significance was witnessed in the phosphorylation of mTOR subsequent to 5101 overexpression and knockdown. As previously stated in the introduction, the activation of the PI3K pathway has the potential to mediate cell autophagy, which could be evaluated using both LC3B and SQSTM1. The findings indicated that the LC3B-II/LC3B-I ratio was elevated by 5101 overexpression and reduced by its knockdown, while no significance was observed in the degradation of SQSTM1 (Appendix A).

To conduct a more in-depth examination of the autophagic regulatory mechanism of 5101 in Kupffer cells under inflammatory conditions, immunofluorescence co-localization analysis was conducted to detect the distribution of LC3B binding to Lyso Tracker Red (Figure 5E). The level of cell autophagy was assessed using both the mean fluorescence intensity of LC3B and the generation of autophagy–lysosome, as outlined in the introduction. Specifically, the mean fluorescence intensity of LC3B was utilized as a metric to evaluate its protein expression level. Simultaneously, the detection of autophagy–lysosome (indicated by the arrows) formation was achieved through the application of the overlap between LC3B and lysosome. An increase in this overlap was found to generate more autophagy–lysosomes, indicating the promotion of cell autophagy. However, the increased overlap also suggested a decrease in the degradative efficiency of autophagy–lysosome, resulting in a retarded autophagic flux conversely. Inhibition of cell autophagy was found to result in a dysfunction of lysosome, thus reducing the degradative efficiency of autophagy–lysosome. This further contributed to the retardation in autophagic flux. The findings indicated that the overexpression of 5101 resulted in a significant increase in the mean fluorescence intensity of LC3B, as measured by Image J, which was consistent with an elevated level of overlap between autophagosomes and lysosomes. This suggested that 5101 accelerated the autophagic flux. Conversely, the knockdown of 5101 led to an increase in the overlap level between autophagosomes and lysosomes, as measured by the Colocalization Finder plugin in Image J, but no statistically significant variance could be observed in the mean fluorescence intensity of LC3B compared to the negative control group, thus indicating that the autophagic flux was intercepted by the knockdown of 5101 (Figure 5F).

In conjunction with the findings from WB and immunofluorescence analyses, it is observed that 5101, in conjunction with 220 as a ceRNA complex, can intensify LPS-induced autophagy in Kupffer cells via the PI3K/AKT/mTOR axis.

### 2.5. Regulation of miRNA 5101 on LPS-Induced Apoptosis in Kupffer Cells via the PI3K/AKT/mTOR Axis

As previously stated in the introduction, CASP3, CASP7, and PARP are comprehensively considered as biomarkers for detecting cell apoptosis. The WB results indicated that the cleavage levels of CASP3, CASP7, and PARP were significantly increased by 5101 (Figure 6A,B), and this effect was reversed upon its knockdown (Figure 6C,D). To confirm the correlation between 5101 and apoptosis in Kupffer cells treated with LPS, both flow cytometry and terminal deoxynucleotidyl transferase-mediated dUTP nick end labeling (TUNEL) assays were employed (Figure 6E–H). The TUNEL assay was accompanied by positive and negative controls to minimize potential systematic errors (Appendix A). The findings from the TUNEL assay indicated that the downregulation of 5101 could mitigate the levels of apoptosis in Kupffer cells that have been treated with LPS (Figure 6E,F). Additionally, the results obtained from flow cytometry demonstrated that 5101 could intensify cell apoptosis in the later stages, but this effect could be reversed by its knockdown (Figure 6G,H).

In summary, the ceRNA complex comprising 5101 and 220, in conjunction with the PI3K/Akt/mTOR axis, plays a central role in modulating the apoptotic levels of Kupffer cells, thereby facilitating their survival during inflammatory conditions.

### 2.6. The Clinical Significance of lncRNA 220

Through the utilization of the NCBI database, a number of homologous sequences of 220 were discovered in humans (Figure 7A), which were subsequently validated in inflammatory cell models employing THP-1 cells (Figure 7B). The miRDB database was employed to predict the target miRNAs of 220 based on its homologous sequences (Figure 7C). Both miR-5100 and miR-3616-3p were identified as relevant miRNAs of 220, and their targeted mRNAs were also predicted using the miRDB database. The findings indicated a close association between miR-5100 (abbreviated as 5100) and miR-3616-3p with the PI3K complex, as evidenced by their targeted mRNAs of PIK3R1 and PTEN, respectively (Figure 7D,E). To further investigate the role of PIK3R1 and PTEN in humans, the GSE54514 dataset from the GPL6947 platform in the GEO database was utilized. The results demonstrated a significant down-regulation of PIK3R1 expressions in peripheral blood cells on the fifth day after admission in non-survivors with sepsis, while no significant differences were observed in survivors with sepsis and healthy individuals (Figure 7F). Concurrently, no statistical significance was observed in the expressions of PTEN in peripheral blood cells on the fifth day in non-survivors with sepsis, survivors with sepsis, and healthy individuals when compared with the first day after admission (Figure 7G). These findings suggested that PIK3R1 played a negative regulatory role in sepsis. Furthermore, the results of immune cell infiltrative analysis in peripheral blood between the first day and the fifth day in non-survivors with sepsis after admission, as determined using CIBERSORTx, indicated that monocytes may be the most relevant cells in sepsis (Figure 7H). The correlative heat map of immune cells (Appendix A) and the correlative lollipop plot between the mRNA expression of PIK3R1 and the infiltrative proportion of various immune cells suggested a strong correlation between PIK3R1 and mast cells, plasma cells, and B cells (Figure 7I). These findings indicated that further research on the regulation of PIK3R1 in sepsis should focus on these specific cell types. Additionally, the ROC diagnostic curves demonstrated that PIK3R1 was a highly significant diagnostic marker for advanced sepsis (Figure 7J), outperforming PTEN (Figure 7K).

As previously stated in the introduction, the PI3K pathway plays a role in multiple diseases through its involvement in cell proliferation and death. The results of this study indicate that 220 may act as a ceRNA complex to regulate the expression of PIK3R1, thus influencing the progression of sepsis. These results underscore the potential value of 220 as a novel target with clinically diagnostic and therapeutic implications for sepsis.

## 3. Discussion

Sepsis is a multifaceted immune disorder syndrome that can arise from trauma, infection, and toxin exposure, leading to immune response dysregulation and organ failure [61]. The pathogenesis of sepsis is linked to the inflammatory response triggered by various pathogen-associated molecules [62]. Given its function in protecting the host against pathogens in the context of sepsis, the liver is considered the most vulnerable organ. Extensive research has demonstrated that septic liver injury increases mortality in sepsis patients, while its amelioration can reverse this outcome [63]. Regrettably, the management of septic liver injury continues to be ineffective as a result of the obscurity of its underlying mechanism, underscoring the pressing need to develop innovative approaches for its diagnosis and treatment.

As indicated in the literature review, lncRNAs are implicated in the pathogenesis of several human disorders, such as sepsis. In order to establish a potential correlation between lncRNAs and sepsis-associated liver injury, systemic inflammated animal models were generated using C57 mice aged 8–12 weeks to mimic the principal pathological events of sepsis, resulting in the frequently observed ALI induced by LPS. Previous studies have suggested that ALI is frequently observed at any stage of sepsis. To advance the exploration of the correlation between lncRNAs and ALI, these models were subjected to LPS treatment at 0 h, 2 h, 8 h, and 24 h, and their hepatic samples were analyzed using RNA-seq. The RNA-seq results demonstrated the active expression of lncRNAs during the 8-h period, with a notable presence of lncRNA 220 (abbreviated as 220), suggesting its potential involvement in the pathogenesis of ALI in endotoxemia in mice. Subsequent analysis revealed a down-regulation of 220 expression following TLR4 and MyD88 knockout, indicating their upstream regulation of 220. Furthermore, tissue-level FISH analysis revealed the predominant expression of 220 in both inflammatory cells and parenchymal hepatic cells. In comparison with the expressions of 220 in AML-12 cells following LPS treatment, a significant disparity was observed in the expressions of Kupffer cells. Furthermore, the results of nucleocytoplasmic RNA separation, in conjunction with FISH analysis at the cellular level, revealed that 220 could be detected in both the nucleus and cytoplasm of Kupffer cells, with a predominant localization in the nucleus, indicating its potential role as a regulator in both of these cellular compartments.

Previous research has highlighted the significant role of the ceRNA complex, composed of lncRNA and miRNA, in regulating genetic expression. To further investigate the regulatory mechanism of 220 concerning the viability of LPS-treated Kupffer cells, the miRDB database was utilized to identify potential miRNAs targeted by 220. Our findings indicate that 220 functions as a decoy of miR-5101 to form a ceRNA complex that regulates cell autophagy and apoptosis via the PI3K/AKT/mTOR axis during inflammatory conditions. Nonetheless, the impact of 5101 on the viability of LPS-activated Kupffer cells remains ambiguous, indicating that these findings may uncover a pivotal function of 5101 in the inflammatory reaction to the binding of 220.

In order to ascertain the impact of 5101 binding to 220 on the inflammatory response, a combination of dual-luciferase reporter and RNA pull-down assays were employed to confirm the binding between 220 and 5101. Unexpectedly, it was observed that the inhibition of 5101 expression was alleviated subsequent to the knockdown of 220, and conversely, the expression of 220 was found to be down-regulated following the overexpression of 5101, while being up-regulated after its knockdown. Furthermore, our investigation revealed that 5101 was capable of increasing the mRNA expressions of TNF-α, IL-1β, and IL-6 in the LPS group, thus exaggerating the injury caused by oxidative stress. Conversely, the suppression of Pik3ca expression was observed subsequent to the overexpression of 5101, which was subsequently reversed upon its knockdown. These results suggested that 220 sponged 5101 as a ceRNA complex to regulate their expressions, thereby mediating the expression of their downstream targets. These findings are of interest and warrant further investigation.

The results obtained from the WB analysis indicated that the promotion of PIK3R1 phosphorylation was observed upon treatment with 5101. However, a reduction in AKT phosphorylation was also observed, leading to the suppression of the PI3K/AKT/mTOR axis. This inhibition could be reversed by the knockdown of 5101. Additionally, overexpression of 5101 resulted in an increase in the LC3B-Ⅱ/LC3B-Ⅰ ratio, which was reversed upon knockdown of 5101. Furthermore, the outcomes of the immunofluorescence co-localization analysis indicated that the average fluorescence intensity of LC3B was heightened in tandem with the levels of overlap between autophagosomes and lysosomes in comparison with the negative control groups. This finding served to demonstrate that the promotion of apoptosis in Kupffer cells, which is triggered by LPS, was facilitated by 5101. Additionally, the levels of cleavage of CASP3, CASP7, and PARP were significantly up-regulated by 5101, and its inhibition was found to effectively reverse these effects. The TUNEL assay demonstrated a decrease in the proportion of TUNEL-positive cells following the knockdown of 5101. Simultaneously, the flow cytometry results indicated that 5101 up-regulated advanced apoptosis in Kupffer cells treated with LPS, an effect that was reversed by its knockdown. Collectively, these findings suggest that 5101, in conjunction with 220, functions as a ceRNA complex to regulate LPS-induced autophagy and apoptosis in Kupffer cells via the PI3K/AKT/mTOR axis.

Subsequently, a number of homologous sequences of 220 in humans were retrieved from the NCBI database, and the targeted miRNAs were screened based on their sequences. The two most pertinent miRNAs of 220 in humans were determined to be miR-5100 and miR-3616-3p, and their downstream targeted mRNAs were identified using Cytoscape. Remarkably, PIK3R1 and PTEN, which exhibited a high degree of correlation with miR-5100 and miR-3616-3p, respectively, were discovered to operate within the PI3K/AKT/mTOR pathway, thereby implying that 220 may serve as a ceRNA complex to sequester miRNAs and to regulate the activity of the PI3K/AKT/mTOR pathway in human beings.

In order to ascertain the impact of PIK3R1 and PTEN on the regulation of moderate to severe sepsis occurrence and progression, the GSE54514 dataset from the GPL6947 platform in the GEO database was utilized. The results were particularly promising, as they revealed a significant decrease in PIK3R1 expression on the fifth day compared with the first day in non-survivors with sepsis after admission. Conversely, no significance was observed in survivors with sepsis or healthy individuals. Concurrently, no statistically significance was witnessed in the expression of PTEN on the fifth day compared to the first day among non-survivors with sepsis, survivors with sepsis, and healthy individuals after admission. Additionally, immune infiltration analysis and ROC diagnostic curve were conducted to demonstrate the significant role of PIK3R1 in the clinical diagnosis and treatment of advanced sepsis. In summary, these findings suggest that prioritizing PIK3R1 as a novel target for future clinical diagnosis and treatment of advanced sepsis in combination with a ceRNA complex composed of 220 and 5100 may be beneficial.

In summary, Figure 8 illustrates the regulatory mechanisms of 220. Upon treatment with LPS, the TLR4-MyD88-dependent pathway serves as the predominant downstream regulatory pathway. The TLR4 receptors located on the membranes of Kupffer cells transmit signals to the nucleus through phosphorylated MyD88, thereby facilitating the transcription of 220. As previously discussed, lncRNAs can be transported between the nucleus and cytoplasm to execute their respective functions in response to external stimuli. A small quantity of 220 is transported to the cytoplasm through the nuclear pores, where it combines with 5101 to create the ceRNA complex. This complex subsequently regulates the downstream PI3K/AKT/mTOR signaling pathway by controlling the levels of phosphorylated PIK3R1, AKT, and mTOR, thereby contributing to the autophagy and apoptosis processes in Kupffer cells during inflammatory conditions. As previously stated in the introduction, extant research has established the crucial role of Kupffer cells in the hepatic and systemic responses to pathogens. These cells are capable of safeguarding the liver against infection during the ensuing inflammatory response by establishing crosstalk with hepatocytes in various hepatic infectious diseases, such as sepsis-associated ALI. However, when the survival state of Kupffer cells is compromised by inflammatory autophagy or apoptosis, their protective effects on the liver are diminished, thereby disrupting the correlation with hepatocytes. This ultimately leads to the compromise of the liver’s immune barrier, resulting in the further development of ALI.

In general, our study utilized RNA-seq to identify a novel lncRNA, named 220, in LPS-induced endotoxemic mice models. We subsequently validated its regulatory effects on autophagy and apoptosis by acting as a ceRNA complex with 5101 in vitro in Kupffer cells. Furthermore, we confirmed the clinical significance of 220 in sepsis by utilizing public databases. Our findings suggest that 220 holds diagnostic and therapeutic potential for sepsis and its associated complications, including ALI. Furthermore, it is imperative to recognize that this study is constrained by specific limitations stemming from the restricted accessibility of experimental resources. Specifically, the absence of primary cell validation, incomplete exploration of the underlying mechanism, and the dearth of in vivo and clinical validations are noteworthy factors. Consequently, further research is indispensable to elucidate these crucial mechanisms in a comprehensive and substantive manner.

## 4. Materials and Methods

### 4.1. Animal Model

LPS (Sigma, L-2880-100 MG, Burlington, MA, USA) was dissolved in phosphate buffered saline (PBS) to achieve a final concentration of 5 mg/mL and subsequently stored at −20 °C for future use. Male C57BL/6 mice, aged 8 to 12 weeks, were procured from the Laboratory Animal Center at Southern Medical University. TLR4^−/−^, MyD88^−/−^, and TRIF^−/−^ mice were generously donated by Prof. T.R. Brilliar from the Department of Surgery at the University of Pittsburgh, Pittsburgh, PA, USA.

The mice were maintained in a specific-pathogen-free (SPF) environment, with regulated temperature and humidity levels. During the feeding process, these mice were exposed to a standard dietary regimen, unrestricted access to potable water, and alternating 12-h periods of light and darkness, while maintaining an air exchange rate of 12–15 times per hour. Prior to the study, the mice were acclimatized for a minimum of one week. To establish LPS-associated endotoxemic animal models, the mice were subjected to anesthesia through inhalation of isoflurane and intraperitoneally injected with a single lethal dose of 20 mg/kg of LPS, as per the relevant literature [64,65]. The animal experiments performed in this study had received approval from the Animal Care and Use Committee of Southern Medical University, Guangzhou, China (L2018235, 17 December 2018).

### 4.2. RNA-Seq and Bioinformatic Analysis

#### 4.2.1. Construction of RNA Library and RNA-Seq

The RNA-seq analysis for both lncRNA and mRNA was conducted by Novogene Bioinformatics Technology (Beijing, China), and the procedure was executed as follows.

C57BL/6 mice were allocated randomly to four groups (*n* = 3 per group) and subsequently intraperitoneally injected with LPS at a single dose of 20 mg/kg at different time points (0 h, 2 h, 8 h, and 24 h) to simulate the initial, intermediate, and advanced phases of sepsis. The pre-cooled Trizol reagent (Invitrogen, 15596026, Waltham, MA, USA) was utilized to extract total RNA from the livers of each group. The assessment of RNA quality was conducted through the utilization of 1.5% agarose gel electrophoresis, while the determination of RNA concentration was accomplished through the application of a NanoDrop Microvolume UV-Vis Spectrophotometer (Thermo Fisher Scientific, Waltham, MA, USA).

Following RNA quality monitoring, the Ribo-Zero rRNA Removal Kit (Epicentre, Madison, WI, USA) was employed to eliminate ribosomal RNA (rRNA) from the total RNA. Subsequently, the NEBNextR µltraTM Directional RNA Library Prep Kit (NEB, Ips-wich, MA, USA) was utilized to append index codes to the attribute sequences of each sample in accordance with the manufacturer’s recommendations, thereby constructing the sequencing libraries. Quality monitoring of the sequencing libraries was conducted using the Qubit 2.0 Fluorometer (Thermo Fisher Scientific, Waltham, MA, USA) for preliminary quantification, followed by precise quantification using qPCR analysis. An Agilent 2100 biological analyzer was utilized to detect the insert size, and upon confirmation of eligibility, sequencing was conducted on the Illumina HiSeq 4000 platform to acquire raw data.

In order to procure uncontaminated data, the raw data underwent filtration by means of a script that eliminated inferior or adapter-contained reads. The sequencing data were assessed based on parameters, such as Q20, Q30, and GC content, and the extent of sequence repetition. Subsequently, the mapped data of each sample were assembled using Stringtie [66] following comparison with the relevant mouse genome database.

The study utilized the CPC, CNCI, CPAT, and PFAM as annotation references to identify transcripts lacking coding potential. Transcripts larger than 200-nt with a minimum of 2 exons were considered as potential lncRNA candidates and were further analyzed using Cuffcompare [67]. The FPKM was calculated using StringTie, taking into account the gene length and gene-mapped read count [68]. The criteria utilized for identifying differentially expressed genes encompassed an adjusted FDR < 0.05 and |log_2_ (Fold change)| > 1. Furthermore, the standardized counts and fragments per kilobase of transcript per million mapped reads (FPKM) were identified as significant factors influencing log2 (Fold change).

The raw sequencing dataset utilized and analyzed in this study can be obtained from the corresponding author upon reasonable request.

#### 4.2.2. Identification of lncRNA 220 and Construction of an Interactive Network between lncRNA–miRNA–mRNA

Utilizing RNA-seq results, volcanic and heat maps were employed to identify 220 entities under the screening condition of |log2 (Fold change)| > 1, FDR < 0.05. R programming language was utilized to perform visualizations for these findings. The characteristics of the 220 entities were authenticated through nucleotide BLAST of the NCBI database and BLAT Search of the UCSC database. The lncLocator 2.0 software (http://www.csbio.sjtu.edu.cn/bioinf/lncLocator2/ (accessed on 2 June 2023)) was applied to predict the subcellular localization of 220.

In order to examine the regulatory effects of 220 on LPS-associated apoptosis and autophagy in Kupffer cells, a network involving lncRNA–miRNA–mRNA interaction was established. The miRDB database was utilized to forecast the miRNAs targeted by 220, as well as their corresponding targeted mRNAs, via the Custom Prediction and Target Search function. Additionally, the String database was employed to identify the associated network of the targeted mRNAs of the 220 targeted miRNAs, based on the outcomes of the miRDB database. The Degree algorithm was utilized to identify hub genes through the Cytohubba plugin in Cytoscape 3.8.2. The functional pathways of these hub genes were predicted using GO/KEGG enrichment analysis, and the interactive network of lncRNA–miRNA–mRNA was visualized using a Sankey plot. Both the GO/KEGG enrichment analysis and Sankey plot were implemented using the R programming language.

#### 4.2.3. Research for the Clinical Significance of lncRNA 220

The nucleotide BLAST function of the NCBI database was employed to identify the homologous sequence of 220 in humans. Additionally, the miRDB database was utilized to predict the targeted miRNAs of 220 in humans and their corresponding targeted mRNAs. The String database was employed to locate the associated network of the targeted mRNAs of the 220-targeted miRNAs, and hub genes were identified using Cytohubba plugin of Cytoscape 3.8.2 under the Degree algorithm. CIBERSORTx was utilized to conduct immune cell infiltrative analysis. The R programming language was applied to visualize the results of immune-infiltrative analysis and ROC curves.

### 4.3. Cell Culture and Treatment

Murine Kupffer cells, obtained from the BeNa Culture Collection (BNCC340733, Beijing, China), were maintained in RPMI-1640 medium (Gibco, C11875500BT, Waltham, MA, USA) supplemented with 10% fetal bovine serum (FBS, Gibco, 10099-141C, Auckland, New Zealand) at 37 °C under a controlled atmosphere of 95% air and 5% CO_2_.

Murine AML-12 cells, procured from the National Collection of Authenticated Cell Cultures (SCSP-550, Beijing, China), were maintained in DMEM/F-12 (1:1) (Gibco, C11330500BT, USA) supplemented with 10% FBS, 1% insulin-transferrin-selenium (ITS) liquid medium supplement (included with F-12), and dexamethasone (Sigma, D4902, USA) at a concentration of 40 ng/mL in the final solution, following the same conditions as Kupffer cells. Both Kupffer cells and AML-12 cells were subjected to LPS treatment at a dose of 100 ng/mL, as per the relevant literature [69].

Hominine THP-1 cells, purchased from the American Type Culture Collection (TIB-202, Manassas, VA, USA), were maintained in RPMI-1640 supplemented with 10% FBS, following the same conditions as Kupffer cells. The cells were then polarized into M0 macrophages by exposure to phorbol 12-myristate 13-acetate (PMA, Sigma, P8139-1MG, USA) at a concentration of 100 ng/mL in the final solution for 24 h [70]. Subsequently, the cells were recultivated in DMEM (Gibco, C11995500BT, USA) containing 10% FBS under the same conditions as before, with a concentration of 100 ng/mL in the final solution of intercalated resistin (Peprotech, 450-19-25, Rocky Hill, NJ, USA) according to the relevant literature [71,72] to induce polarization into M1 macrophages.

### 4.4. Cell Transfection

Small interfering RNA of 220 (si-220) and its corresponding negative control (si-NC) were synthesized by GenePharma (Shanghai, China), with a final intervening concentration of 100 pmol/mL. Mimics of 5101 (miR-5101) and its corresponding negative control (miR-NC), as well as the inhibitor of 5101 (anti-miR-5101) and its corresponding negative control (anti-NC) were wholly synthesized by RiboBio (Guangzhou, China). The administered concentrations of miR-5101 and miR-NC were established at 100 pmol/mL, while those of anti-miR-5101 and anti-NC were doubled. The transfection process was executed through the utilization of Lipofectamine 3000 (Invitrogen, Waltham, MA, USA) once the cells had achieved a confluence of 70–80%. The mRNA expression level was detected at the 36-h mark post-transfection, whereas the protein level was detected at the 48-h mark post-transfection.

### 4.5. Quantitative Real-Time PCR (qRT-PCR)

The Kupffer cells underwent lysis through the utilization of pre-cooled Trizol on ice for a duration of 5 min, followed by the extraction of their total RNA via chloroform. The RNAs were then precipitated with isopropanol and resuspended in 75% pre-cooled ethanol. The RNA concentration was assessed by means of NanoDrop Microvolume UV-Vis Spectrophotometer (Thermo Fisher Scientific, Waltham, MA, USA) after dissolution in diethyl pyrocarbonate (DEPC)-treated water. Reverse transcription was performed using the ReverTra ACE qRT-PCR RT Kit (TOYOBO, FSQ-101, Tokyo, Japan), and the resulting cDNA was quantified using Power Green qRT-PCR Mix (Dongsheng Biotech, P2102, Guangzhou, China) under 2^−ΔΔCt^ method. The primer sequences are given in Appendix A. Ultimately, 18S, β-actin and U6 were wholly selected as the internal controls.

### 4.6. RNA Pull-Down Assay

Following the acquisition of the complete sequence of 220, a biotin-labeled long non-coding RNA (lncRNA 220) probe (Bio-lncRNA 220) and a control probe (Bio-Gapdh) were synthesized via in vitro transcription utilizing T7 RNA Polymerase (Roche, 10881767001, Basel, Switzerland). To isolate the targeted RNA from the total RNAs, M-280 Streptavidin Dynabeads (Invitrogen, USA) were employed with 20 μg of the total RNAs and incubated with the aforementioned probes at 4 °C for over 16 h, respectively. Subsequently, quantitative analysis was conducted using qRT-PCR using the primer sequences provided in Appendix A.

### 4.7. RNA Fluorescence In Situ Hybridization (FISH) and Nucleocytoplasmic RNA Separation Assays

A fluorescence hybridization probe coupled with CY3 was synthesized by GenePharma to detect the distribution of 220 at the tissue and cell level, following their established protocols. The Cytoplasmic and Nuclear RNA Purification Kit (NORGEN, 37400, Mississauga, ON, Canada) was utilized to separate the nucleocytoplasmic RNAs of LPS-treated Kupffer cells, and the quantification was performed through qRT-PCR analysis.

### 4.8. Western Blot Assay

Proteins were extracted utilizing pre-cooled RIPA buffer (GenStar, E121-01, Shanghai, China) and performed to a quantification using the BCA Protein Assay Kit (Solarbio, PC0020, Beijing, China). Following separation via polyacrylamide gel, the proteins were transferred onto polyvinylidene difluoride (PVDF) membranes and subsequently subjected to blocking with either 5% skimmed milk or 5% bovine serum albumin (BSA, Sigma, USA). The primary antibodies were then applied at a proportion of 1:1000 and incubated with the membranes overnight at 4 °C, with the exception of ACTB which was applied at a proportion of 1:5000. Following an overnight incubation, the membranes were subsequently subjected to probing with secondary antibodies conjugated with HRP at a proportion of 1:5000 for a minimum of 60 min at ambient temperature. Subsequently, visualization was accomplished by employing the ECL luminescence reagent (Biosharp, BL520A, Hefei, China).

### 4.9. Dual-Luciferase Reporter Assay

The 3′UTR sequence of 220 was inserted into Psicheck2 vectors to construct the 220-WT reporters, which contained the binding sites of 5101. The mutant sequence was utilized for the construction of the 220-MUT reporters. Following co-transfection of the two reporters and mimics of 5101 or its negative control into Kupffer cells, the quantification of luciferase activity was performed through the utilization of the Dual-Glo Luciferase Assay System Kit (Promega, E2920, Fitchburg, WI, USA).

### 4.10. Flow Cytometry

Following various transfections, the Kupffer cells were subjected to digestion utilizing 0.25% Trypsin without EDTA. The Annexin V-FITC/PI Apoptosis Kit (Elabscience, E-CK-A211, Wuhan, China) was employed to assess the apoptotic status of the cells, as per the guidelines provided by the manufacturer.

### 4.11. Immunofluorescence Assay

To assess the autophagy flux of Kupffer cells following various treatments, co-localization analysis between autophagosomes and autolysosomes was conducted using Alexa Fluor 488 (Abcam, ab150077, Cambridge, UK) and Lyso Tracker Red (Solarbio, L8010, China). Upon completion of transfection, Lyso Tracker Red was introduced to serum-free medium at a ratio of 1:20,000 and incubated with the cells at 37 °C under 5% CO_2_ conditions. After incubation, the cells were fixed using 4% paraformaldehyde for 30 min at room temperature, and subsequently permeabilized using 0.5% Triton X-100 for 15 min under ambient conditions. Following the conditions of the experimental setting, the cells were subjected to a 30-min blockage with 3% BSA (prepared with DPBS). Whereafter, they were incubated with a LC3B specific antibody (Cell Signaling Technology, 12741, Danvers, MA, USA) at a ratio of 1:100, maintained at 4 °C overnight. Following this, the cells were subjected to an incubation with Alexa Fluor 488 at a ratio of 1:1000 for a period of 1 h at ambient temperature under light-restricted conditions. Eventually, the cells underwent counterstaining with 4′ 6-diamidino-2-phenylindole (DAPI) for a duration of 20 min under the same experimental conditions, followed by imaging via confocal microscopy.

### 4.12. TUNEL Assay

The One-step TUNEL In Situ Apoptosis Kit (Elabscience, E-CK-A325, China) was utilized to identify the existence of apoptotic cells. The TUNEL reagents were employed to stain the apoptotic cells in accordance with the protocol and subsequently observed under a fluorescence microscope.

### 4.13. Hematoxylin and Eosin (H&E) Staining

The liver tissue underwent fixation with a 4% paraformaldehyde solution for a minimum of 24 h. Following this, the tissues were embedded in paraffin and sectioned into slices measuring 3-μm in thickness, which were subsequently subjected to staining with hematoxylin and eosin. Eventually, the stained sections were sealed with liquid sealing and scrutinized under a light microscope.

### 4.14. Statistical Analysis

The experimental data are presented as means ± SEM from a minimum of three independent experiments, utilizing the SPSS Statistic 23 software. The statistical significance threshold of *p* < 0.05 is utilized to analyze the differences within each group through the implementation of one-way ANOVA, two-way ANOVA, and *t*-test.

## 5. Conclusions

In summary, the findings suggest that 220 is significant in the modulation of LPS-induced autophagy and apoptosis in Kupffer cells through its interaction with 5101 as a ceRNA complex via the PI3K/AKT/mTOR axis, thereby mediating the onset of LPS-induced ALI. Furthermore, 220 can serve as a reasonable therapeutic target for the clinical management of sepsis by acting as a decoy for 5100 as a ceRNA complex via the PI3K/AKT/mTOR axis.

## Figures and Tables

**Figure 1 ijms-24-11210-f001:**
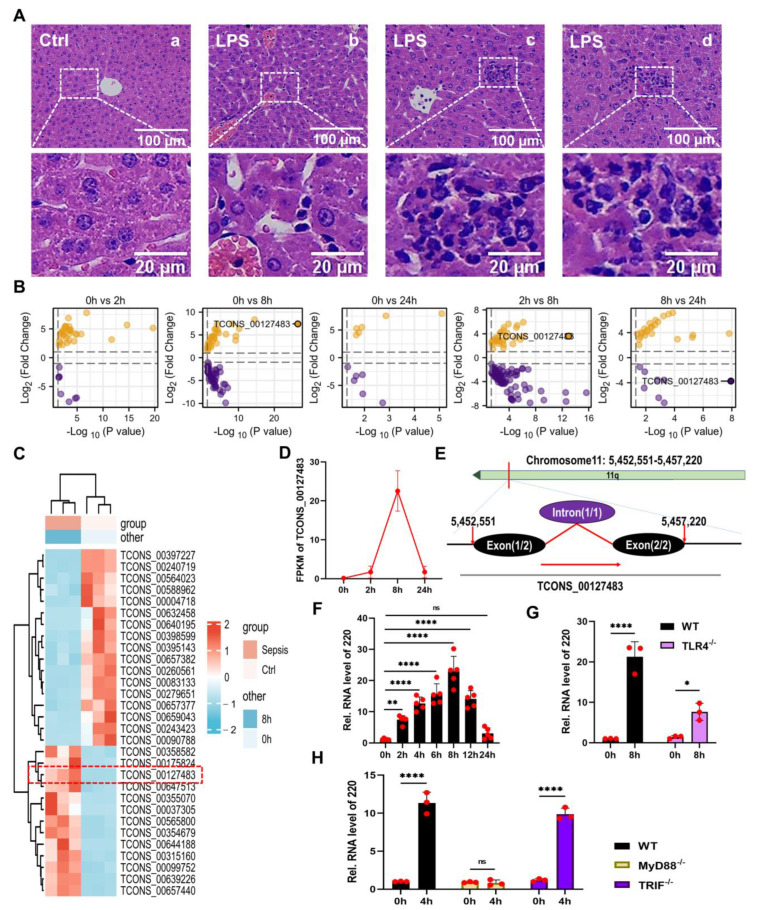
Identification and annotation of lncRNA 220. (**A**) H&E staining for the livers of mice treated with LPS. (**B**) Volcano plots generated to illustrate the differentially expressive lncRNAs in the livers of mice treated with LPS at different time points (0 h, 2 h, 8 h, and 24 h). (**C**) Heat map of differentially expressive lncRNAs in the livers of mice treated with LPS for 8 h (the lncRNA exhibiting the highest degree of differential expression is depicted within the red-dashed enclosure). (**D**) Expressive trend plot of lncRNA TCONS_00127483. (**E**) Chromosomal location of lncRNA TCONS_00127483 in the mouse genome. (**F**) Expressions of 220 in the livers of mice treated with LPS at different time points (0 h, 2 h, 4 h, 6 h, 8 h, 12 h, and 24 h). (**G**) Expressions of 220 in the livers of WT and TLR4^−/−^ mice treated with LPS for 8 h. (**H**) Expressions of 220 in the livers of WT and MyD88^−/−^, TRIF^−/−^ mice treated with LPS for 4 h (*, *p* < 0.05, **, *p* < 0.01; ****, *p* < 0.0001; ns, not significant).

**Figure 2 ijms-24-11210-f002:**
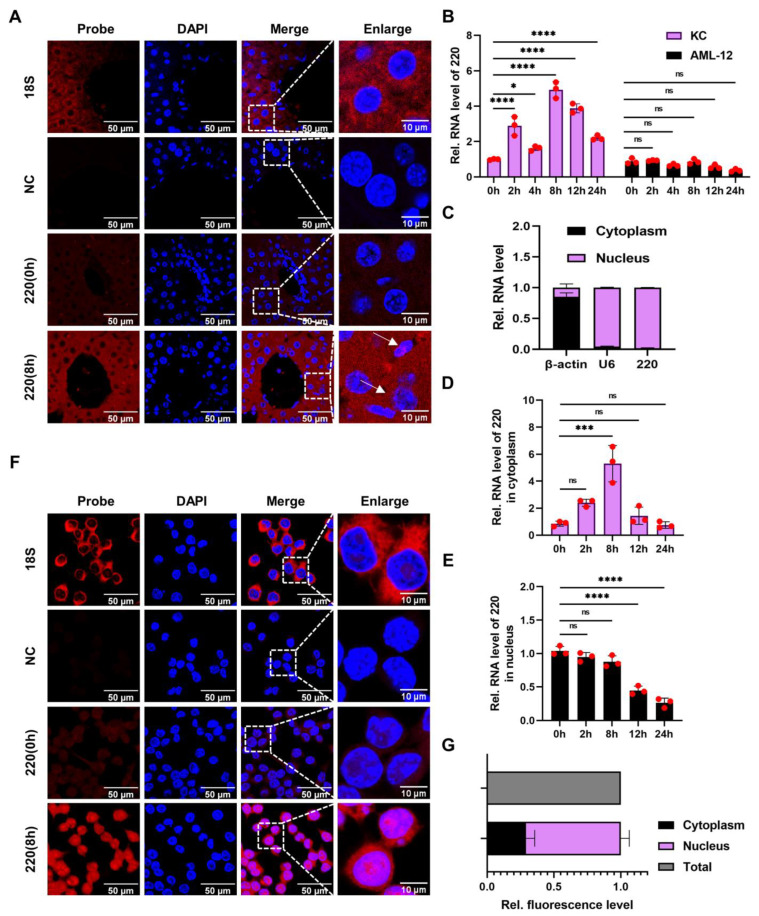
Identification and annotation of lncRNA 220. (**A**) FISH assay utilized to examine the distribution of 220 at the tissue level. (**B**) Expressions of 220 in Kupffer cells and AML-12 cells treated with LPS at different time points (0 h, 2 h, 4 h, 8 h, 12 h, and 24 h). (**C**) Nucleocytoplasmic distribution of 220 in Kupffer cells at the normal condition. (**D**) Variable trend of cytoplasmic expressions of 220 in Kupffer cells treated with LPS at different time points (0 h, 2 h, 8 h, 12 h, and 24 h). (**E**) Variable trend of nuclear expressions of 220 in Kupffer cells treated with LPS at different time points (0 h, 2 h, 8 h, 12 h, and 24 h). (**F**,**G**) FISH assay utilized to examine the distribution of 220 at the cellular level (*, *p* < 0.05; ***, *p* < 0.001; ****, *p* < 0.0001; ns, not significant).

**Figure 3 ijms-24-11210-f003:**
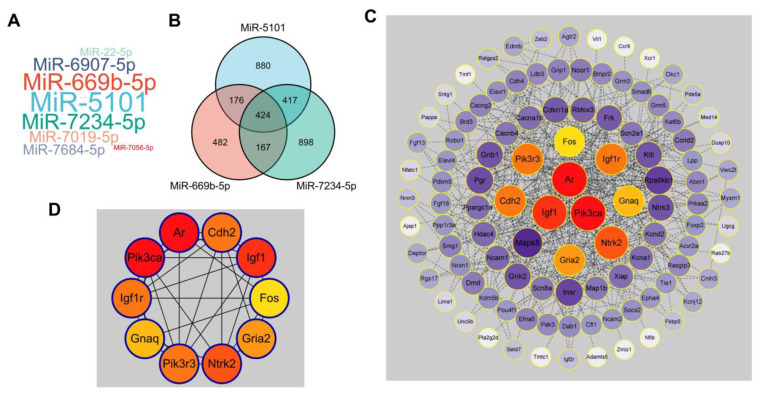
Prediction of the ceRNA complex constructed by lncRNA 220 and its targeted miRNAs. (**A**) Prediction of the targeted miRNAs of 220. (**B**) Venn plot of the overlapping downstream mRNA targets of miR-5101, miR-669b-5p, and miR-7234-5p. (**C**) Interactive plot of downstream targeted mRNAs of 5101. (**D**) Interactive plot of 10 hub targeted mRNAs of 5101 screened out by Cytoscape (the hue of the icon symbolizes the significance of genes, with a deeper red hue indicating a greater degree of importance). (**E**) GO/KEGG enrichment analysis for the hub targeted mRNAs (the manifestation of the hub targeted mRNAs’ participation in the PI3K pathway is exhibited within the red-dashed enclosure). (**F**) Sankey plot of ceRNA complex formed by 220 and 5101 (the color used in this plot is primarily employed for the purpose of designating each variable and its interrelation). (**G**) Prediction of the regulatory pathway involving the interaction of 220 and 5101.

**Figure 4 ijms-24-11210-f004:**
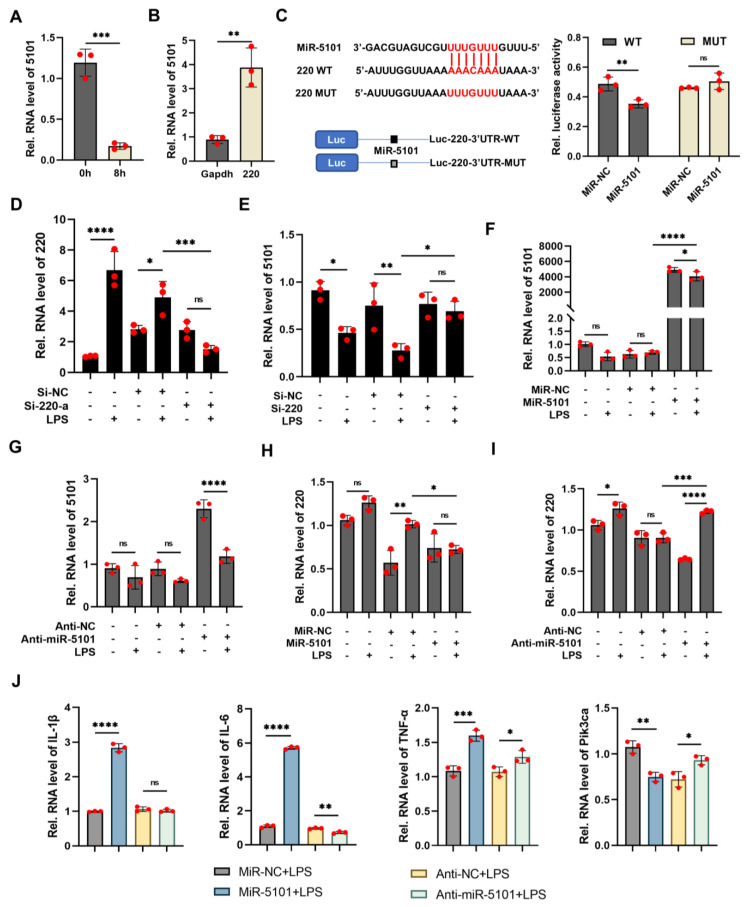
Decoy of miRNA 5101 by lncRNA 220 as a ceRNA complex. (**A**) Expressions of 5101 in the livers of mice treated with LPS for 8 h. (**B**) RNA pull-down assay conducted to confirm the interaction between 220 and 5101. (**C**) Dual-luciferase reporter assay conducted to confirm the interaction between 220 and 5101 (the complementary interaction sequence between 220 WT and 5101, as well as the mutation sequence of the corresponding 220 MUT are exhibited in the red-fonts box). (**D**) Interfering efficiency of 220. (**E**) Expressions of 5101 after knockdown of 220. (**F**) Overexpression efficiency of 5101. (**G**) Interfering efficiency of 5101. (**H**) Expressions of 220 after overexpression 5101. (**I**) Expressions of 220 after knockdown of 5101. (**J**) Assessment for the mRNA levels of inflammatory cytokines and Pik3ca after overexpression and knockdown of 5101 (*, *p* < 0.05; **, *p* < 0.01; ***, *p* < 0.001; ****, *p* < 0.0001; ns, not significant).

**Figure 5 ijms-24-11210-f005:**
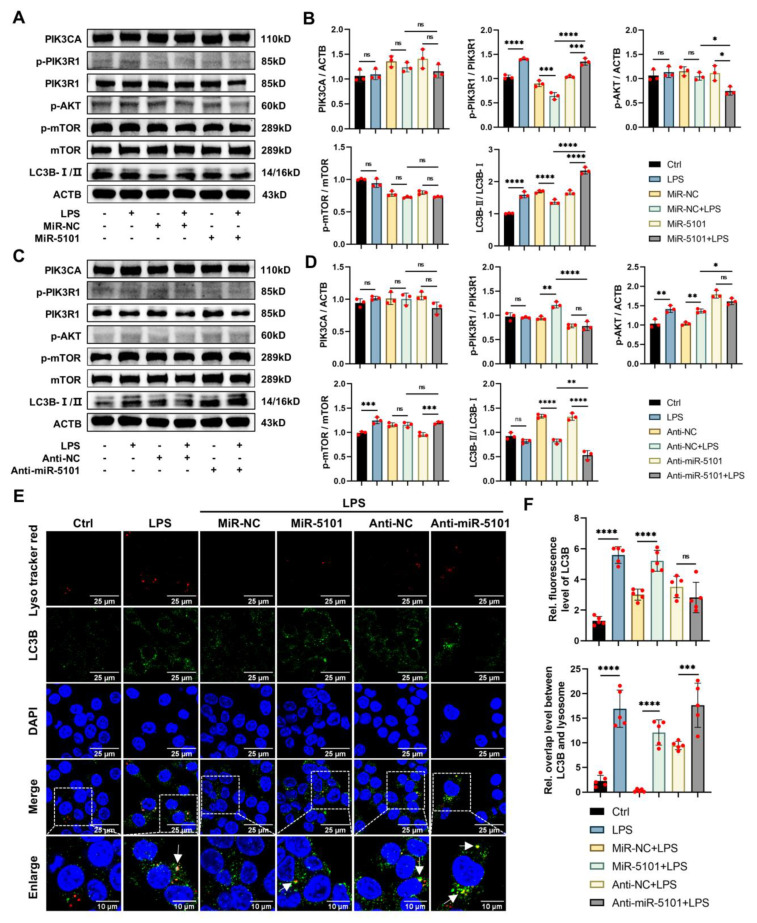
Up-regulation of miRNA 5101 on LPS-induced autophagy in Kupffer cells. (**A**,**B**) WB results pertaining to the expressive levels of pertinent proteins within the PI3K/AKT/mTOR signaling pathway and cell autophagy process after overexpression of 5101. (**C**,**D**) WB results pertaining to the expressive levels of pertinent proteins within the PI3K/AKT/mTOR signaling pathway and cell autophagy process after knockdown of 5101. (**E**,**F**) Immunofluorescence co-localization assay conducted to detect the autophagic flux after overexpression and knockdown of 5101 (*, *p* < 0.05; **, *p* < 0.01; ***, *p* < 0.001; ****, *p* < 0.0001; ns, not significant).

**Figure 6 ijms-24-11210-f006:**
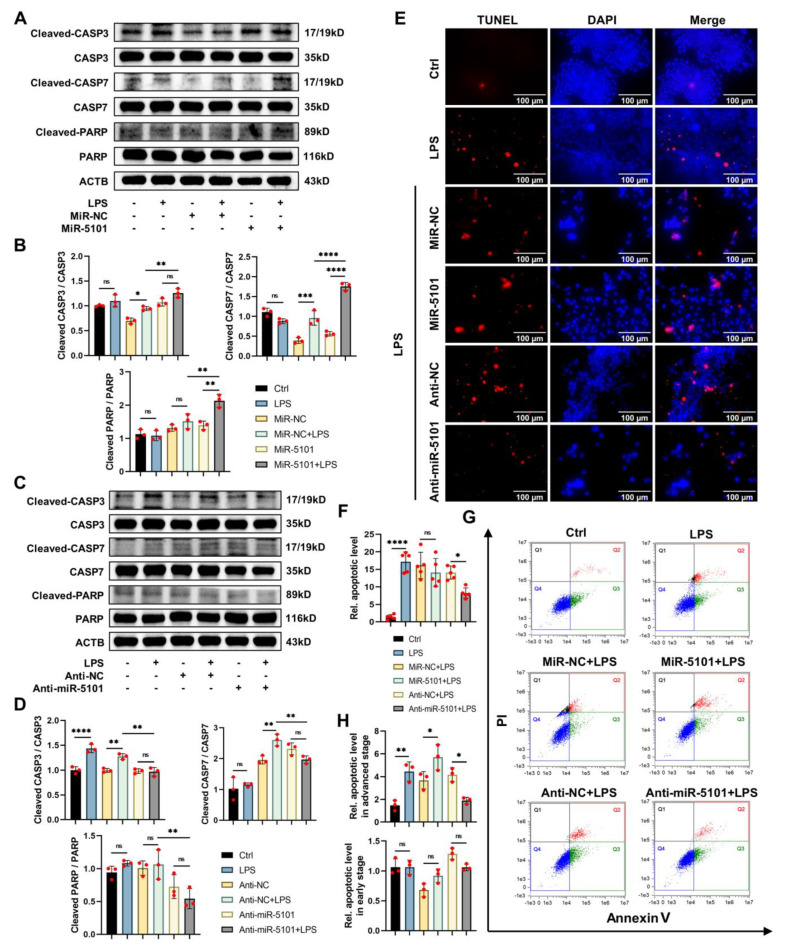
Up-regulation of miRNA 5101 on LPS-induced apoptosis in Kupffer cells. (**A**,**B**) WB results pertaining to the expressive levels of pertinent proteins of cell apoptosis process after overexpression of 5101. (**C**,**D**) WB results pertaining to the expressive levels of pertinent proteins of cell apoptosis process after knockdown of 5101. (**E**,**F**) TUNEL assay after overexpression and knockdown of 5101. (**G**,**H**) Flow cytometry after overexpression and knockdown of 5101 (*, *p* < 0.05; **, *p* < 0.01; ***, *p* < 0.001; ****, *p* < 0.0001; ns, not significant).

**Figure 7 ijms-24-11210-f007:**
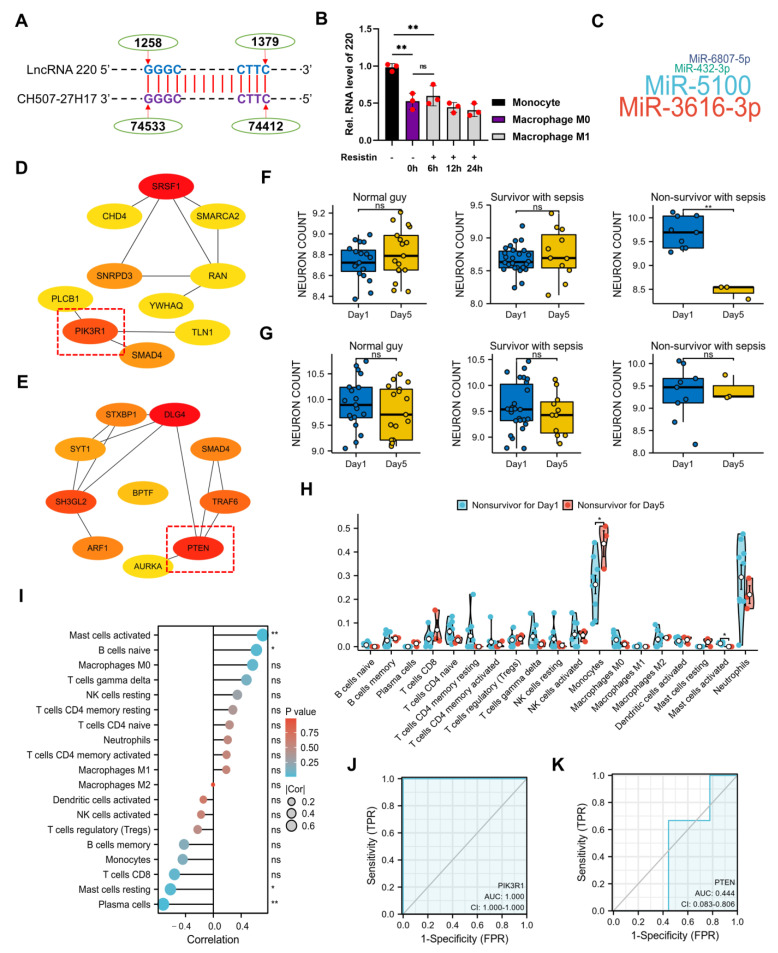
The clinical significance of lncRNA 220. (**A**) Homologous sequences of 220 on chromosome 21 in human. (**B**) Validation of the expressions of 220 in human cells. (**C**) Prediction of the downstream targeted miRNAs of 220 in human. (**D**) Correlation between PIK3R1 (highlighted within the red-dashed box) screened out by Cytoscape and miR-5100 (the hue of the icon symbolizes the significance of genes, with a deeper red hue indicating a greater degree of importance). (**E**) Correlation between PTEN (highlighted within the red-dashed box) screened out by Cytoscape and miR-3616-3p. (**F**) Expressions of PIK3R1 on the fifth day compared to the first day after admission in normal guys, survivors, and non-survivors with sepsis, respectively. (**G**) Expressions of PTEN on the fifth day compared to the first day after admission in normal guys, survivors, and non-survivors with sepsis, respectively. (**H**) Immune cell infiltrative analysis in peripheral blood conducted on non-survivors with sepsis after admission, comparing the first day to the fifth day. (**I**) Correlative lollipop plot between the mRNA expression of PIK3R1 and the infiltrative proportions of various immune cells. (**J**) ROC diagnostic curve of PIK3R1 in non-survivors with sepsis. (**K**) ROC diagnostic curve of PTEN in non-survivors with sepsis (*, *p* < 0.05; **, *p* < 0.01; ns, not significant).

**Figure 8 ijms-24-11210-f008:**
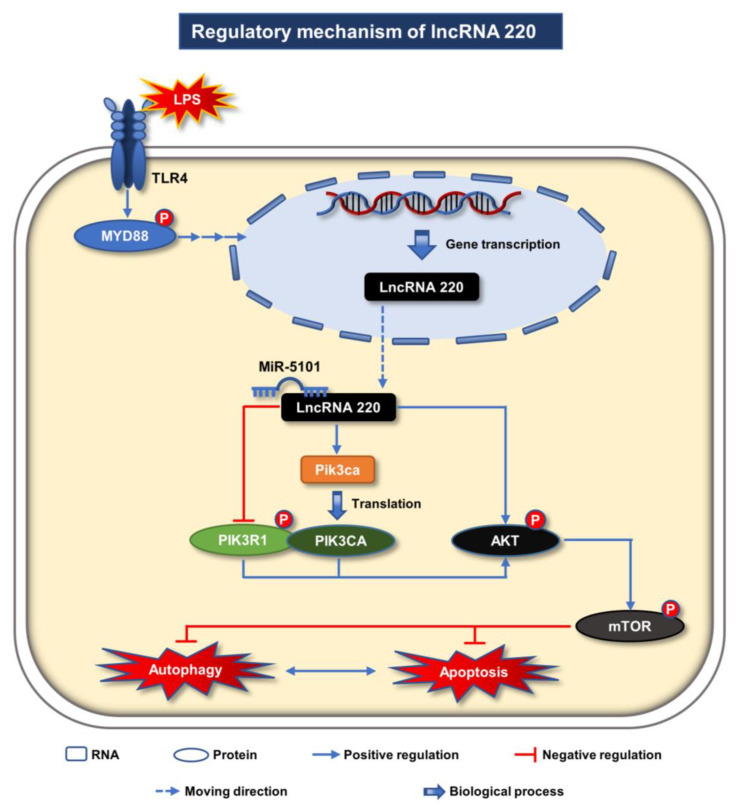
Regulatory mechanism of lncRNA 220. Upon LPS treatment, the TLR4-MyD88 dependent pathway functions as the primary downstream regulatory pathway. The TLR4 receptors situated on the Kupffer cell membranes transmit signals to the nucleus via phosphorylated MyD88, thereby promoting the transcription of 220. A minor proportion of 220 is transported through the nuclear pores to the cytoplasm, where it complexes with 5101 to form the ceRNA complex. During inflammatory conditions, the complex is responsible for regulating the downstream PI3K/AKT/mTOR signaling pathway by modulating the levels of phosphorylated PIK3R1, AKT, and mTOR, which in turn contributes to the autophagy and apoptosis processes in Kupffer cells.

## Data Availability

The raw sequencing dataset supporting the results of this study can be obtained from the corresponding author upon reasonable request. The clinical dataset utilized in this study is available on the GEO platform under the accession number GSE54514, using the GPL6947 platform (https://www.ncbi.nlm.nih.gov/geo/query/acc.cgi?acc=GSE54514 (accessed on 2 May 2023)).

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
