# Peer review of "LncRNA 220: A Novel Long Non-Coding RNA Regulates Autophagy and Apoptosis in Kupffer Cells via the miR-5101/PI3K/AKT/mTOR Axis in LPS-Induced Endotoxemic Liver Injury in Mice"

_ijms, 2023, doi:10.3390/ijms241311210_

Round 1

Reviewer 1 Report

The abstract outlines some background including an introduction to sepsis and the role of non-coding RNA. Then lncRNA 220 is introduced, and the regulatory loop forms with miR-5010 and is linked to ALI by mediating autophagy and apoptosis.

The introduction sets out to introduce sepsis, its various characteristics and the current gap in knowledge. However, the authors could add a sentence after reference 5 to outline what the current therapies are and that they are ineffective/ not very effective so that the next sentence makes more sense.

Be known to all; this is an unusual expression and I haven’t seen it used before in scientific text, consider changing it.

The authors then introduce ALI as a facet of sepsis and that it is linked to poor prognosis. Then the authors mention general characteristics of lncRNAs, miRNAs, and PI3K/AKT pathways but they could also add a reference to link these to sepsis/ ALI.

Some links about the signalling cascades linked to sepsis/ ALI, especially those mentioned in this study are required. For example, TLR4 and MYD88 get referred to later and the authors assume prior knowledge by the readers (at any rate, the authors need to justify why these genes and their mutants were included in the work even if all readers know about TLR4 and MYD88/ NK-kB/ TRIF/ RAGE.  This is the main weakness of the intro not explaining the signalling and immunity pathways. 

In the next paragraph, the authors build momentum about previous studies they have conducted on the ALI mouse model and the RNA-seq analyses which generated a list of non-coding RNA linked to sepsis and that the current study will investigate this non-coding RNA. However, if this is the case, your previous study must be cited. Also, in the paragraph (60-77), I am not which aspects belong to the previous work and which ones to the current work. This section here needs rewriting and better clarification so that current aims/ objectives are stated clearly and plainly.

Methods:

In line 404, how was LPS administered and how many doses were given?

In section 4.2. if the authors have generated the RNA-seq dataset, there needs to be a lot more detail about the exact methods used, bioinformatic analyses, and where the data is currently stored for public use. If the data is not generated by these authors, then kindly mention where it originates from and what the accession numbers are. The various tools mentioned such as string, Cytoscape and the like settings used need to be explicitly mentioned. Just mentioning the authors used there will in no shape or form assist reproducibility.

Please conduct thorough English language editing.

In 4.3. if the authors have bought various cell lines from companies, the information about this must be detailed. Also, AML-12/ THP-1 etc, owned by the lab, needs to have been sourced from somewhere, either a gift or purchase, please detail this.

Results:

2.1. again, please specify in this was a single dose of LPS. Also, If there is any useful information about animal husbandry and daily inspection/ maintenance (e.g., weight and health checks) available to assist repeat of your experiments by readers, this would be good.

In Figure 1A, in the magnified version of H&E sections in A, please specify what magnification and if new scale bars are relevant, please add them.

(C) Heat map of differential expressive lncRNAs in the livers of mice stimulated by LPS for 0 h and 8 h. Can the authors specify why these specific lncRNAs were shown and what does the whole heatmap (for all lncRNAs) look like? The authors could add this information (if available) to the supplementary.

From there on the authors refer to the lncRNA as 220 and look at its expression in the mouse model. In F, can the authors speculate as to why the level of lncRNA-220 has gone down after 12 hours?

The authors have not mentioned by TLR4/ MYD88 -/- models were used, the link will be apparent to some readers but not to all.

G-H need justification in the results and also an introduction in the introduction section. The readers remain unaware of why these KO mice models have been used.

Also, pertinent to my last comment, the authors need to specifically list every single mouse model used in this study in the methods and material and also provide due citation or acknowledgement of where each line originated from. I have rechecked the methods and this information does not appear.

Figure 2:

I can’t tell from the labelling if the data in 2A and D has been quantified, please provide image feature quantifications and add this to the figure. E seems to quantify something but I’m not if it is quantifying 2A and 2D.

In B-C there doesn’t seem to be a discernible trend for lncRNA-220 expression post incremental LPS levels. How do the authors explain this? Also, the authors could justify why the nuclear/ cytoplasmic is so unpredictable. They do mention that this lncRNA might be a regulator but what exactly does it regulate? Also, if it is a cytoplasmic regulator with nuclear presence, why would this then give a similar expression pattern across the samples? Since molecules will rapidly get shuttled between the nucleus and cytoplasm and this lncRNA has a presence in both why would the ratio change erratically by LPS treatment? Also, why did the authors drop 24-hour data?

Figure 3:

The authors describe bioinformatic analyses to pinpoint the miRNA targeted of lncRNA-220. Also, they were associated (showed by STRING). In C, please define the functional modules Cytoscape has identified. The link between C-D needs to be made clearer. How was this link formed and what is its implication? Interestingly, these genes in D are linked to PI3K. 

Also, the Sankey plot in f needs to be explained more. What exactly does it show and signify?

Figure 4: B pull-down assay showing the association between the miRNA and lncRNA and where the binding would occur in C. Convincing data in C showing that miR-5101 binds lncRNA-220 3’UTR.  D, the authors report that the two molecules don’t form strong interactions. In E, the LPS treatment leads to reduced miR5010 levels but the loss of 220 nullifies this effect. The authors show that 220 is degraded by 5101.

I would suggest talking about cytokines in J in the intro where you were meant to expand on signalling. Convincing data about miRNA5101 upregulated some cytokines but inhibited pik3ca.

In 5 the authors provide some convincing WB data about the link between 5101 and PIK3R1 or 5101 and LC3II/I. Also, the WB data has been quantified as well which is great. The quality of the p-PIK3R1 blot is not great and this is a crucial finding if the authors have a better representative then that would be great. 

I forgot to say earlier, the authors are referring to various autophagy proteins and players, please define these also in the introduction. Ironically, the word autophagy does not appear once in the intro, please rectify this.

LC3B goes up in the presence of LPS when miR5101 is present and goes down when anti-miR5101 is used. In line 206, the label is F and not D.  Please also explain the significance of the overlap between LC3B and lysosomes and a readout (none of this has been explained at all). One of many of these signalling pathways and autophagy processes and players, the authors assume prior knowledge from all readers.  Even for the sake of argument if all readers knew these markers well, still the authors need to provide some explanation and justification as to why these readouts for used.

Figure 6: Starting a whole new paragraph with however is not grammatically or logically correct. Since it indicates some prior comparison.

The authors find that various apoptosis mediators such as initiator caspases alter due to 5101 OE and KD. Do the authors, however, have better representative images for the cleaved caspases in A and C? I appreciate these might be tricky antibodies to work with and that perhaps the blots were stripped and reblocked several times. For G please use a white background for the flow cytometry plots to make it easier to discern what percentage of cells lie in each quadrant (I appreciate early and late apoptosis, Q3/ Q2, have been quantified).

Simultaneously, the results of flow cytometry demonstrated that 5101 could exacerbate the cell apoptosis in an advanced stage, whereas the observation could be reversed by its knockdown (Figure 6G, H).  Why do you think this was the case?

Overall, the authors show that apoptosis is elevated by 5101.

Figure 7. This figure uses various databases to show the importance of 220, which is good.

Figure 8 is useful.

Some questions remain. 

The authors in Figure 4 state that the interaction between 5101 and 220 was weak, so if this miRNA is not inducing its effect through repressing 220 (by Watson-crick complementation/ binding), then could it be repressing PIK3R1 and subsequently AKT/ mTOR through an independent mechanism?

But the phosphorylated level of PIK3R1 was elevated after overexpression of 5101, whereas its knockdown reversed this result. 5101 also suppressed the phosphorylated level of AKT, whereas this result could be reversed by its knockdown.  Since miRNAs can bind to mRNA, is it possible that 5101 is directly binding to PIK3R1 or p-AKT and regulating them?

Also, the links found between 5101 and 220 (and the latter's sponging effect) are useful but it is also remaining is an integrative view of all this. For example, how does 5101/ 220 affect target autophagy and apoptosis to mediate the ALI/sepsis phenotype? How are autophagy and apoptosis themselves forming the general outline of sepsis/ ALI?  In Figure 1A, the authors show necrotic tissue and don’t develop hypotheses about what logistically apoptosis and autophagy are doing to deliver sepsis/ ALI or what specific contributions are they making. 

In my opinion, the molecule phenotype and consequence of the interaction of 5101/220 has been developed quite well with many experiments and good controls but through a weak introduction and aims section, the actual question mark and what the authors ultimately wanted to show is still unclear.

I would remodel the question asked by including more detail about the molecular basis of sepsis, its links to apoptosis and autophagy plus PI3K/AKT pathways and then making clear the aims.

Since this information is largely unclear the following statement remains confusing:

Taken together, the conclusions could be drawn as follows: first, 220 modulates autophagy and apoptosis in Kupffer cells in the septic liver injury by decoying 5101 as a ceRNA complex via the PI3K/AKT/mTOR axis; next, 220 could be considered as a novel target for the clinical diagnosis and treatment of advanced sepsis by decoying 5100 as a ceRNA complex to mediate the function of the PI3K/AKT/mTOR axis. 

Why would mediating PI3K/AKT/mTOR helps the management of sepsis? This is why I stated in the introduction that the non-coding RNA and PI3K/AKT/mTOR were not linked to sepsis.

Many thanks

Some English editing is required.

Reviewer 2 Report

Thank you for giving me the opportunity to review this article.

The authors investigated the sepsis-related gene expression changes focusing on lncRNA and miRNA, especially lnc 220 and miR-5101. Although the results of this study are based on experiments in mice, they appear to be new insights into altered gene expression in sepsis.

I think the contents of this article are interesting to readers; however, some minor modifications or additional comments are needed.

The following is my comment.

-The authors described, "The expression of 5101 was elevated in the septic animal models stimulated by LPS for 8 h (Figure 4A)" on page 7, line 156. But in Figure 4A, the 5101 expression level seems to be decreased.

Round 2

Reviewer 1 Report

The authors have addressed all of my comments and have made substantial changes to their manuscript. Thanks.

A little proof-reading would be beneficial
